# Pancreatic Cystic Lesions: A Focused Review on Cyst Clinicopathological Features and Advanced Diagnostics

**DOI:** 10.3390/diagnostics13010065

**Published:** 2022-12-26

**Authors:** Wei Chen, Nehaal Ahmed, Somashekar G. Krishna

**Affiliations:** 1Department of Pathology, The Ohio State University Wexner Medical Center, Columbus, OH 43210, USA; 2School of Medicine, The Ohio State University, Columbus, OH 43210, USA; 3Division of Gastroenterology, Department of Internal Medicine, The Ohio State University Wexner Medical Center, Columbus, OH 43210, USA

**Keywords:** pancreatic cystic lesions, morphology, endomicroscopy, histology, molecular, NGS, nCLE, advanced diagnostics

## Abstract

Macroscopic, endomicroscopic, and histologic findings and correlation are an integral part of the diagnostic evaluation of pancreatic cystic lesions (PCLs), as complementing morphologic features seen by different specialties are combined to contribute to a final diagnosis. However, malignancy risk stratification of PCLs with worrisome features can still be challenging even after endoscopic ultrasound guided-fine needle aspiration (EUS-FNA) with cytological evaluation. This review aims to summarize cyst clinicopathological features from the pathologists’ perspective, coupled with knowledge from advanced diagnostics–confocal laser endomicroscopy and cyst fluid molecular analysis, to demonstrate the state-of-art risk stratification of PCLs. This review includes illustrative photos of surgical specimens, endomicroscopic and histologic images, and a summary of cyst fluid molecular markers.

## 1. Introduction

Pancreatic cystic lesions (PCLs) are being detected more frequently due to the increasing use of cross-sectional imaging [1]. Our knowledge of different types of PCLs has grown dramatically in recent years. A multidisciplinary approach using a combination of radiological imaging, endoscopic ultrasound (EUS), cytology, cyst fluid analysis, and molecular profiling is most helpful in diagnosing and risk stratifying PCLs [2,3,4,5]. Despite these advancements, challenges still exist in the appropriate classification of PCLs as benign versus precancerous and accurate risk-stratification of intraductal papillary mucinous neoplasm (IPMNs). Unwarranted resection of benign PCLs (up to 15% of pancreatectomies) causes unnecessary mortality and morbidity [6], and there is continued surgical overtreatment of branch duct-IPMNs where up to 50% of lesions only reveal low-grade dysplasia [7]. 

Various guidelines and recommendations exist for the management of PCLs [8]. The authors’ institution follows the 2017 international consensus guidelines for intraductal papillary mucinous neoplasm (IPMN) of the pancreas [1]. While the differentiation of most PCLs can be achieved through a combination of the clinical history, gender, imaging characteristics, cytology and cyst fluid analyses, malignancy risk stratification of PCLs with worrisome features can still be challenging even after EUS-FNA with cytological evaluation. This review aims to summarize cyst clinicopathological features from the pathologists’ perspective, coupled with knowledge from advanced diagnostics—confocal laser endomicroscopy and cyst fluid molecular analysis—to demonstrate the state-of-art risk stratification of PCLs.

EUS-guided needle-based confocal laser endomicroscopy (nCLE) generates real-time endomicroscopic images of the PCLs. Coupled with intravascularly injected fluorescein dye, nCLE accentuates the vascular pattern of the lesion revealing distinct epithelial features. Benefiting from a preserved interstitium, in vivo nCLE captures certain vascular features less obvious on histology (example: serous cystadenoma) and can increase the accuracy of differentiation of PCLs. Molecular analysis by next-generation sequencing (NGS) has evolved significantly in recent years with improved accuracy in the diagnosis and risk-stratification of PCLs [9]. 

According to recent surgical series of PCLs [10,11,12], the most resected PCLs are mucinous neoplasms: IPMNs (45% of all resected PCLs) and mucinous cystic neoplasm (MCN, 16%). The most frequently resected non-mucinous neoplasms include serous cystic neoplasms (SCN, 16%), cystic neuroendocrine tumors (cNET, 5%), and solid pseudopapillary neoplasm (SPN, 3%) [13]. In this review, for each of these and other commonly encountered PCLs, the definition, epidemiology, location, macroscopic, endomicroscopic, histologic features, and molecular alterations will be reviewed systematically to help with the differential considerations and the ultimate construction of the final diagnosis. 

## 2. Intraductal Papillary Mucinous Neoplasm 

**Definition:** Intraductal papillary mucinous neoplasm (IPMN) of the pancreas is a grossly visible (>5 mm) intraductal epithelial neoplasm arising in the main duct (MD) and/or branch-duct (BD) of the pancreas. The epithelia of IPMN feature papillary formation and mucin production. 

**Epidemiology:** Peak incidence is between 62–67 years of age. In the USA, MD-IPMN is slightly more common in men (male:female ratio 1.1:1), while BD-IPMN is more common in women (male:female ratio 0.76:1) [14].

**Location:** IPMN can occur anywhere in the pancreatic ductal system, with the head of the pancreas being the most common location. Multicentricity (synchronous or metachronous lesions) is observed in up to 40% of cases; therefore, clinical follow-up of the remnant pancreas is important even if the resection margin is negative [15,16,17,18,19,20]. 

**Macroscopic appearance:** MD-IPMNs are mostly located in the head of the pancreas. If the more proximal pancreatic duct is involved, mucin may be seen extruding from the dilated ampullary orifice into the duodenum (fish-mouth papilla) [21,22]. The main duct in IPMN is by definition >5 mm in diameter and filled with tenacious mucin and lined by friable papillary formations [17]. If secondary ducts are also involved, it is classified as mixed duct-type IPMN. 

BD-IPMNs mostly occur in the uncinate process and present as peripheral multilocular cystic lesions. A gross photo of a resected BD-IPMN is shown in Figure 1A–D. The cut surface of the pancreas demonstrates characteristic multilocular, Swiss cheese appearance. Some cysts have smooth lining (Figure 1A gross photo short arrow; Figure 1B histology), while others are lined by papillary neoplastic proliferation (Figure 1A gross photo long arrow; Figure 1C histology). The adjacent pancreatic parenchyma often undergoes lobular atrophy (Figure 1D between arrows). 

IPMNs > 3 cm and/or with mural nodules/papillary projection are likely to harbor high-grade dysplasia and invasive carcinoma. A MD-IPMN is shown (Figure 1E,F). The cyst is lined by neoplastic epithelium and the lumen contains papillary/villous growth, partially obstructing the duct lumen (Figure 1E). This case contains both low-grade dysplasia (Figure 1F upper half) and high-grade dysplasia (Figure 1F lower half). 

**Endomicroscopy:** EUS-nCLE of IPMN shows finger-like papillary projections composed of an outer epithelium and inner vascular core [23]. Figure 2 shows the typical appearance of IPMNs as seen during real-time in vivo EUS-nCLE procedure. Figure 2A shows a BD-IPMN with low-grade dysplasia. The epithelium is thinner and more translucent. Comparatively, Figure 2B demonstrates BD-IPMN with high-grade dysplasia with thicker and darker epithelium indicative of cellular and nuclear stratification, respectively. Both computer-aided artificial intelligence algorithms and human-interobserver studies have revealed a high diagnostic accuracy in nCLE-guided differentiation of IPMNs [24,25].

### Histopathology

**Histologic Subtypes:** There are three histologic subtypes of IPMN, based on the predominant cell differentiation of the neoplastic epithelium [14]. The former oncocytic subtype is now considered a separate entity “Intraductal Oncocytic Papillary Neoplasm (IOPN)”, due to its different molecular underpinning and prognosis [26,27,28,29]. 

The three histologic subtypes of IPMNs are gastric type (~70%), intestinal type (~20%), and pancreatobiliary type (~10%) (Figure 3). Gastric type (Figure 3A) resembles gastric foveolar epithelium, composed of tall columnar cells with basally oriented small nuclei and abundant cytoplasmic mucin. Scattered goblet cells may be seen. Gastric type is often associated with low-grade dysplasia and usually occurs in branch ducts [30]. Intestinal type (Figure 3B) is the second most common histologic type, which most often resides in the main duct and is associated with high-grade dysplasia. It features hyperchromatic villous papillae, containing elongated and crowded nuclei, basophilic cytoplasm and variable amount of goblet cells [31,32,33,34]. Pancreaticobiliary type (Figure 3C) contains amphophilic to eosinophilic cytoplasm, enlarged nuclei with nucleoli, and moderate cytoplasmic mucin. It typically involves the main duct and often shows high-grade cytoarchitectural atypia. Some consider the pancreatobiliary type a variant of high-grade gastric type [31,35]. 

**Grading of dysplasia:** IPMNs is currently classified as low grade and high grade, based on the highest degree of cytoarchitectural atypia in the epithelium [14,36]. Low-grade IPMNs are characterized by mucinous epithelium with mild to moderate atypia. The epithelium may be flat (Figure 3A), show small tufting, or arrange into tall papillae but with bland cytology (Figure 4A). High-grade IPMNs feature severe atypia, characterized by papillae with irregular branching and budding, nuclear stratification with loss of polarity, cellular pleomorphism, and increased mitoses (Figure 4B–D). Notably, with increasing dysplasia, the papillae may lose the central fibrovascular core. In Figure 4B, while the bottom villous structure still has a fibrovascular core, the villi in the center of the image become slender without stromal core and the nuclei are rounded up, showing high-grade features including pleomorphism and loss of polarity. 

**Molecular alterations:** Somatic mutations in *KRAS* and *GNAS* are the two most common genetic alterations seen in IPMN, together seen in >95% of all IPMNs [37,38]. *KRAS* mutation is an early event and nearly a prerequisite in the pathogenesis of ductal neoplasms, including pancreatic ductal adenocarcinomas (PDAC). Activating *GNAS* mutations are enriched in the intestinal subtype [39,40,41]. *RNF43* somatic mutations are seen in about 50% of IPMNs [37,42]. Changes in tumor suppressor genes, such as *TP53, CDKN2A* and *SMAD4*, as well as mutations of mTOR genes (*PTEN*, *PIK3CA, AKT1*) are associated with advanced neoplasia (high-grade dysplasia and invasion) [43,44,45,46,47,48,49]. Alterations of other genes in the mitogen-activated protein kinase (MAPK) pathway have also been detected in mucinous cysts, including *BRAF*, *ERBB2*, *HRAS*, and *MAPK1*. *GNAS* and *BRAF* mutations in mucinous cysts are fairly specific for IPMN [9,46].

**IPMN-associated invasive carcinoma:** Overall, about a third of resected IPMN are associated with invasive carcinoma. Main-duct IPMN (>10 mm) has higher rate of associated malignancy, up to 60%, whereas branch-duct IPMN (>3 cm) has a lower rate (15–20%) [50,51,52,53,54]. The two most common IPMN-associated invasive carcinomas are tubular (ductal) adenocarcinoma and colloid adenocarcinoma. Colloid adenocarcinomas almost always arise in a background of intestinal-type IPMN, typically harbor GNAS mutations, and carry a better prognosis. Tubular adenocarcinoma is morphologically similar to conventional pancreatic ductal adenocarcinoma (PDAC) and is associated with pancreaticobiliary/gastric-type IPMN, and KRAS mutation [40,55,56]. The invasive component can be very focal, therefore thorough (if not complete) sampling of the resected pancreas is warranted.

## 3. Mucinous Cystic Neoplasm 

**Definition:** Mucinous cystic neoplasm (MCN) of the pancreas is a mucin-producing cystic epithelial neoplasm that is associated with characteristic ovarian-type subepithelial stroma. 

**Epidemiology:** The mean patient age is 48 years. MCN predominantly (>98%) occurs in women [14]. Increasing age correlates with a higher risk of invasive carcinoma, suggesting that progression occurs over a period of years. 

**Location:** Almost exclusively, MCNs occur in the body or tail of the pancreas (>98%) [57,58,59,60]. Unlike IPMN, MCN is typically solitary, without synchronous lesions.

**Macroscopic appearance:** The mean size of MCN is 6 cm (ranging 2 to 35 cm). MCNs with invasive carcinoma are larger (>5 cm, mean: 9 cm) [57]. MCNs can be unilocular or multilocular with a few septa. They do not communicate with the pancreatic ductal system and typically have a thick fibrous wall (≥3 mm, Figure 5A top arrow) [61]. The pathogenesis is unclear and thought to be related to ectopic embryonic remnants, which explains some “ancient” changes such as a thick fibrous wall and mural calcifications. High-grade MCNs often have mural nodules or papillary projections.

**Endomicroscopy:** EUS-nCLE of MCNs shows horizon-type epithelial bands (Figure 6A) of variable thickness without papillary conformation [23]. The epithelial bands are single or in multiple layers and are best observed when viewed tangentially to the nCLE probe. Further, MCNs can reveal clusters of inflammatory cells with areas of a dark background and fluorescent macrophages. Since MCNs may contain foci of atrophy and chronic inflammation, nCLE characterization of epithelial bands can sometimes be challenging during in vivo EUS. Inflammatory clusters of cells or debris can also be observed in pseudocysts, but such patients frequently have a history of pancreatitis and nCLE does not reveal any epithelial bands (Figure 6B,C). 

**Histopathology:** The mucinous epithelium of MCNs is morphologically similar to that of IPMNs, and the dysplasia is also graded based on a two-tiered system [36]. It is not uncommon for the epithelium of MCN to be denuded in areas, sometimes extensively. Looking for the subepithelial ovarian-type stroma is helpful in such cases (Figure 5B arrows), and the presence of this distinctive stroma is required for the diagnosis of MCN [14]. However, the ovarian-type stroma may be attenuated (hypocellular/hyalinized) in large MCNs, postmenopausal patients, or around areas with advanced neoplasia [57]. Positive immunohistochemical staining for progesterone receptor (60–90%) (Figure 5C) and estrogen receptor (30%) may aid in the detection of the ovarian-type stroma [60,62].

Invasive carcinoma typically occurs in MCNs that are large (>5 cm) or with gross papillary nodules [60,63,64]. Invasion is seen in about 15% of MCNs [57,59,65]. Similar to IPMNs, the invasive component is usually tubular/ductal adenocarcinoma and can be focal, so extensive/complete sampling of the cyst wall is indicated [14,57,66].

**Molecular alterations:** Activating mutations in codon 12 of *KRAS* is seen in 50–66% of MCNs as well as loss of function in *RNF43* [9,41,67]. Unlike IPMN, *GNAS* mutations are rarely seen in MCN [43]. The gene mutations associated with advanced neoplasia in IPMN may be observed in MCN, including *TP53, CDKN2A*, *SMAD4*, and/or mTOR genes (*PTEN*, *PIK3CA, AKT1*) [9].

## 4. Serous Cystadenoma

**Definition:** Serous cystadenoma (SCA) is a benign epithelial neoplasm composed of uniform cuboidal, glycogen-rich clear cells that often form cysts containing serous fluid. 

**Epidemiology:** The mean age at presentation is 58 years, with a female predominance (female: male = 3:1).

**Location:** SCA can occur anywhere in the pancreas; mostly (50–75%) in the body or tail. 

**Macroscopic appearance:** SCAs are well-circumscribed, multilocular cystic mass, without communication with the pancreatic ductal system. The mean size is about 4 cm with a wide range (1 to 25 cm in diameter) [68,69,70,71]. The cyst wall is typically thin (<3 mm) [61,72]. The microcystic variant shows characteristic honeycomb or sponge pattern (Figure 5D,E). A central scar with a sunburst calcification pattern is present in 30% of cases. There are also rare macrocystic (oligocystic) variant and solid variant. 

**Endomicroscopy:** EUS-nCLE of SCA shows an intricate fern pattern of vascularity (Figure 6D,E) or also called superficial vascular network. This interstitial vascular pattern reveals parallel or inter-connected network of capillaries underneath the epithelium [23]. Trafficking red blood cells are frequently observed in the fine vascular meshwork during the in vivo EUS-nCLE procedure. There is variation in contrast (fluorescein) penetration within the capillaries; often, the capillaries are densely packed with red blood cells.

**Histopathology:** The lining epithelium of SCA consists of a single layer of low cuboidal epithelial cells with clear cytoplasm, due to abundant intracytoplasmic glycogen. Variably prominent capillary network is seen underneath the epithelium (Figure 5F). Nuclear atypia and mitoses are typically absent. The neoplastic epithelium is immunoreactive for inhibin, which can be helpful in small biopsy or cytology specimens. Microforceps biopsy may show very scant cuboidal epithelial cells in a background of blood (Figure 5G,H); positive inhibin staining helps to confirm the diagnosis (Figure 5I) in the appropriate clinical setting. 

**Molecular alternations:** Germline or somatic alterations of the tumor suppressor gene *VHL* is present in SCAs [9,41,67,73]. In the setting of germline *VHL* mutation, there could be multifocal SCAs in the pancreas. Alterations in genes associated with IPMN, MCN, and PDAC, such as *KRAS*, *GNAS*, *CDKN2A*, and *SMAD4*, have not been reported in SCAs [41]. *TP53* or *TERT* promoter mutations may be prognostically important, as they are associated with interval growth of cyst size [9].

## 5. Cystic Neuroendocrine Tumor

Pancreatic neuroendocrine tumors are typically well-circumscribed, solid tumors. However, hemorrhage and secondary degeneration may occur in the center of the tumor (Figure 7A,B), resulting in a grossly cystic neuroendocrine tumor (cNET) with only viable tumor cells in the cyst wall. EUS-nCLE of cNET shows dark clusters (trabeculae) of cells separated by stroma [23]. The clustering of cells can conform to various shapes, commonly in cords or groups with occasional geometric formations (Figure 6F). Histologically, cNET is composed of nests, trabeculae, or ribbons of neoplastic cells, separated by thin vascular fibrous septa (Figure 7C). *MEN1* alterations are highly specific for cNETs, but the sensitivity is low (27%) [9]. Loss of *ATRX/DAXX* and the presence of alternative lengthening of telomeres (ALT) are associated with poor prognosis [9]. cNETs with loss of heterozygosity (LOH) of ≥3 genes tend to have distant metastasis [9].

## 6. Solid Pseudopapillary Neoplasm

**Definition:** Solid pseudopapillary neoplasm (SPN) is a low-grade malignant tumor that lacks a specific line of pancreatic epithelial differentiation. It most likely arises from genital ridge cells that were translocated to the pancreas during embryogenesis [74]. 

**Epidemiology:** SPNs occur predominantly (90%) in young women. The mean age at presentation is 28 years.

**Location:** SPNs have a slight preference for the pancreatic tail [75,76]. 

**Macroscopic appearance:** SPNs are well-demarcated mass lesions, with variable solid and cystic components, and occasional calcifications. SPNs are typically large tumors (average size: 8 cm; range: 0.5–25.0 cm) [14]. The cystic component reflects degenerative changes secondary to hemorrhagic necrosis (Figure 7D). Not surprisingly, small tumors tend to be more solid. 

**Endomicroscopy:** EUS-nCLE of SPN is indistinguishable from cNET revealing groups or clusters of cells separated by interstitial spaces [23]. 

**Histopathology:** SPN is composed of epithelioid cells, forming pseudopapillary structures due to perivascular growth (Figure 7E) and also crowded nests imparting a solid appearance in areas (Figure 7F). SPNs may mimic a pancreatic NET, macroscopically, endomicroscopically, and histologically. Diffuse nuclear staining of β-catenin, as well as the expression of SOX11 and TFE3 by immunohistochemistry, are helpful to confirm the diagnosis of SPN [77,78,79,80]. Of note, SPNs may be positive for synaptophysin by immunohistochemistry, but chromogranin should be negative.

**Molecular alternations:** Somatic activating mutation in *CTNNB1* (encoding β-catenin) is the main molecular feature of SPNs [41,81,82]. 

## 7. Pseudocyst

A pseudocyst is a collection of fluid contents walled off by fibrous tissue after episode(s) of pancreatitis (Figure 7G). The location is usually outside the pancreas. EUS-nCLE of pseudocyst (Figure 6B,C) reveals a dark background due to the absence of vascular interstitium and a true epithelium-lined cyst wall. Some auto-fluorescent inflammatory cells are usually present in pseudocysts [23]. Histologically, a cyst lining is absent, and the cyst wall is composed of fibroinflammatory tissue, often with evidence of old hemorrhage and necrosis (cholesterol clefts and pigments, Figure 7H). 

## 8. Squamous Lined Epithelial Cysts

**Definition:** Squamous lined epithelial cysts can be divided into three types: lymphoepithelial cysts (LEC), dermoid cysts (cystic teratomas), and epidermoid cysts in intrapancreatic accessory spleen (ECIPAS) [83]. LECs are benign, true cysts lined by mature, keratinizing squamous epithelium with abundant surrounding lymphoid tissue [83,84,85]. They are theorized to arise from misplaced branchial cleft cysts during embryogenesis [86], squamous metaplasia of obstructed intrapancreatic ducts, or inclusion of benign epithelium or ectopic pancreas into a peripancreatic lymph node [85,87]. Dermoid cysts of the pancreas, like other teratomas, are tumors composed of tissue derived from germ layers (ectoderm, mesoderm, and endoderm) [88]. ECIPAS is a cystic mass arising from intrapancreatic accessory spleen (congenital abnormality characterized by ectopic splenic tissue) [89].

**Epidemiology:** LECs are the most prevalent among squamous-lined epithelial cysts and account for approximately 0.5% of all PCLs. The mean patient age at presentation is 56 years, with a male predominance (80%) [84]. Dermoid cysts of the pancreas occur in a younger age group with a mean age of 23 (range 2–53 years) with no gender predominance [83]. The mean patient age for ECIPAS is 38 years [83] with a slight female predominance [90]. 

**Location:** LEC can occur anywhere in the pancreas (head, body, or tail) or can be found in an extra-pancreatic location [83,84,91,92]. Dermoid cysts can occur anywhere along the pathway of ectodermal cell migration and can been found in anywhere in the pancreas [93,94]. ECIPAS are primarily located in the tail of the pancreas [83,90]. 

**Macroscopic Appearance:** The mean size of LECs is approximately 5 cm (range 1.2–17 cm). They are often round and well-demarcated from the surrounding pancreas. They can be multilocular (60%) or unilocular (40%) in appearance. The contents of the cysts appear “cheesy” or “caseous” (signifying keratinaceous debris) or may less often be clear and serous [84]. Dermoid cysts of the pancreas contain a combination of both cystic and solid components. Ectodermal differentiation (skin, hair follicles, sweat glands, sebaceous material) is the most common, but structures from other germ layers (cartilage, bone, thyroid tissue, etc.) may be present [88]. The mean size of ECIPAS is 4.5 cm (range 2.3–6.5 cm). ECIPAS can be unilocular or multilocular. It may contain serous fluid but will lack presence of hair or skin appendages [83,95,96]. 

**Endomicroscopy:** LECs can reveal clusters of bright particles suggestive of keratinous debris otherwise can be heterogenous in appearance with a bland background (squamous epithelium) and lattice-type blood vessels [23,97]. ECIPAS can reveal cords of cells suggestive of ectopic splenic tissue [97].

**Histopathology:** LECs are lined by squamous epithelium and surrounded by subepithelial lymphoid tissue with germinal centers [84,98]. Other rarer features of LECs include: containing sebaceous differentiation, keratin debris and cholesterol crystals, scattered mucinous cells with features similar to goblet cells, and multinucleated giant cells [84,98,99,100,101]. Dermoid cysts can include mucinous epithelium, respiratory-type mucosa, sebaceous units, and hair. Unlike LEC, dermoid cysts lack subepithelial lymphoid tissue [83,84,93]. ECIPAS are typically lined with squamous (keratinized or nonkeratinized) epithelium surrounded by normal splenic tissue [96], although lining containing flattened cuboidal epithelium has been reported [90]. They have been reported to contain blood, cholesterol clefts, macrophages, and microcalcifications [90,96]. 

**Cyst Fluid:** Occasionally cyst fluid from LECs and rarely ECIPAS has elevated carcinoembryonic antigen (CEA) levels causing challenges with differentiation from mucinous neoplasms [84,90,91,95,101].

## 9. Simple Mucinous Cyst

In the absence of a visible obstructive process, mucinous cysts that are >1 cm, without any features of IPMN or MCN, are classified as simple mucinous cysts [36,102,103,104,105]. Cyst fluid analysis may show elevated CEA levels. Histologically, the mucinous epithelium is flat with rare tufting, but well-formed papillae are absent. The most common mutations include *KMT2C* (62%), *KRAS* and *TP53* (15%) [52,106]. However, in 31% cases, no mutations are detected. A simple mucinous cyst is essentially a diagnosis of exclusion, and the main differential considerations include IPMN, MCN, and a retention cyst involved by pancreatic intraepithelial neoplasia. 

Other rare pancreatic cystic lesions not discussed in this review include intraductal oncocytic neoplasm (IOPN), intraductal tubulopapillary neoplasm (ITPN), acinar cystic transformation of the pancreas, among others [13,52,107].

## 10. Advanced and Emerging Diagnostic Tools for Pancreatic Cystic Lesions 

Despite our ever-increasing knowledge of PCLs, there remains continued resection of benign cysts at high rates and resection of IPMNs with low-grade dysplasia. In this review, we have described features observed in specific PCLs using two advanced diagnostics (nCLE and cyst fluid NGS). Both techniques can enhance the diagnostic accuracy of cyst type and advanced neoplasia significantly [108], and they may be considered when radiomics and EUS cyst morphology are indicative of a high-risk PCL [2]. However, these advanced diagnostics are only available in limited centers currently, and large prospective studies are needed before incorporation into guidelines. Other emerging diagnostic tools for PCLs include Mass Spectrometry and Optical Coherence Tomography (OCT). 

Needle-based confocal laser endomicroscopy (nCLE) provides real time, en-face (perpendicular to histologic sectioning planes) visualization of cyst lining at the microscopic level. Through a combined assessment of interstitial vascular pattern, cyst content fluorescence characteristics, and the “negative” image/architecture of the cyst epithelium, the accuracy of nCLE diagnosis of PCLs reaches over 90% [109,110]. In addition, nCLE evaluation of the “thickness” and “darkness” of the cyst epithelia allows grading of dysplasia and thus risk stratification of IPMNs [24]. However, the lack of adequate exposure/training for the real-time interpretation of images impede the widespread use of nCLE. Artificial intelligence (AI)-aided interpretation of nCLE images may help overcome some of the challenges [25]. 

Next generation sequencing of the nucleic acid present in the cyst fluid, offers molecular insight into the genetic makeup of the cells shed from the neoplastic cyst epithelium. Certain genetic alternations indicate specific pancreatic cyst types, and molecular evolution with additional gene mutations is correlated with neoplastic progression and advanced neoplasia [9]. Alternations involving *KRAS*, *GNAS*, and/or *BRAF* indicate mucinous cyst; *GNAS* and *BRAF* mutations in mucinous cysts are fairly specific for IPMN (rare in MCN). Advanced neoplasia in mucinous cysts is associated with alterations in *TP53*, *SMAD4*, *CTNNB1, CDKN2A*, and/or the mTOR genes (*PTEN*, *PIK3CA*, *AKT1*). Cystic pancreatic NETs are associated with *MEN1* mutations and LOH for multiple genes. Loss of *ATRX*/*DAXX* and the presence of alternative lengthening of telomeres (ALT) are associated with poor prognosis in pancreatic NETs. SCAs often harbor *VHL* mutations; *TP53* and *TERT* promoter mutations are seen in large-sized SCAs. 

EUS through-the-needle biopsy (TTNB) represents a “bite” biopsy of the cyst wall by a microforceps introduced through a standard EUS 19-gauge FNA needle. TTNB generates larger tissue fragments and boasts good diagnostic yield (69.5%) and high histological accuracy (86.7%) in a meta-analysis of 454 patients [111]. Another meta-analysis demonstrates that TTNB has a higher sensitivity and specificity than cytology [112]. However, a relatively high rate of complications has been reported with mucinous cysts or cysts with connection to pancreatic duct. In a prospective study of 101 patients who underwent TTNB, adverse event rate was 9.9% (10 patients, including 9 acute pancreatitis and 1 fatality) [113]. Therefore, TTNB should be considered for patients where the necessity of an accurate diagnosis outweighs the risks [114]. We exclusively reserve TTNB for SCAs. Only when nCLE shows SCA (or non-mucinous) pattern, we will consider TTNB.

Mass spectrometry is a useful tool that measures the mass-to-charge ratio of ions and allows for the identification of the molecular weight, structure, and chemical formula of pure substances [115,116]. Additionally, mass spectrometry has been a helpful noninvasive tool that can assist in the early diagnosis of cancer by identifying both the presence of cancer and its progression over time [117,118]. Early mass spectrometry studies analyzing protein profiles obtained from pancreatic cyst fluid found unique peptide patterns for various benign PCLs [119,120], detected pancreatic tumor markers (such as Mucin family members, S100 proteins, CEA-related proteins) [121], and were able to distinguish between benign and malignant pancreatic lesions [120]. Notably, one study combined a novel biomarker panel of four proteins with CA19-9 and was able to diagnose pancreatic carcinoma with a sensitivity of 95% and specificity of 94.1% [122]. In a more recent 2018 diagnostic study, targeted mass spectrometry taken from pancreatic cyst fluid using protein biomarkers from mucin-5AC and prostate stem-cell antigen was able to identify advanced neoplasia with an accuracy of 96%. Targeted mass spectrometry can detect 95% of malignant/severely dysplastic lesions, compared with 35% and 50% for CEA and cytology respectively. Additionally, a panel of peptides taken from mucin-5AC and mucin-2 was able to discriminate between premalignant/malignant and benign lesions with an accuracy of 97%, outperforming cyst fluid CEA (61%) and cytology (84%) [123].

Optical Coherence Tomography is an imaging modality that features a spatial resolution at the µm level. OCT imaging has been applied to the examination of human eyes [124], skin [125], and cardiovascular tissue [126]. Recently, a prototype endoscopic OCT has been reported for early diagnosis and Endo-OCT image-guided brachytherapy of pancreatic cancer and precursor lesions [127]. 

## 11. Summary

Unlike pancreatic cancers that have known risk factors such as smoking, chronic pancreatitis, diet, and diabetes, most neoplastic pancreatic cysts do not have a clear etiology. Perhaps, they may be considered genetic diseases, caused by inherited (germline) and somatic mutations. Patients with certain hereditary cancer syndromes are known to be predisposed to developing pancreatic cysts. IPMNs have been reported in patients with McCune–Albright syndrome, Peutz–Jeghers syndrome, Lynch syndrome, familial adenomatous polyposis, hereditary breast and ovarian cancer syndrome, familial atypical multiple mole melanoma, and Carney complex [128]. SCNs develop in 90% of patients with von Hippel–Lindau syndrome (VHL). Overall, 10–20% of PanNETs are associated with hereditary syndromes including multiple endocrine neoplasia type 1, VHL, neurofibromatosis type 1, tuberous sclerosis, and others [14]. It is unclear why MCNs and SPNs have a female predominance. They could be related to persistent fetal periductal mesenchyme or translocated embryonic tissue from genital ridges that respond and proliferate in response to hormonal stimulation [14]. 

Pancreatic cystic lesions have distinct clinicopathologic, endomicroscopic, and molecular features facilitating differential diagnosis and risk stratification (Table 1). However, there are overlaps and it remains a clinical challenge in the pre-surgical evaluation of PCLs. An integrated multidisciplinary approach is the most effective approach in managing PCLs. This allows sharing of unique insights from different specialties, ultimately fitting the pieces of the puzzle together. This review compares the clinicopathological, endomicroscopic, and molecular features of the most common PCLs, bridging information across specialties with an emphasis on macroscopic, (endo)microscopic morphology, and molecular findings. In the diagnostic workup of a new PCL, demographic information, clinical presentation, and location, can generate a working differential. Distinct image-guided (MRI/CT, and EUS) cyst morphological features (size, locularity, wall thickness, mural nodule, duct communication) can refine and narrow down the differential. Standard of care analysis of the cyst fluid (CEA, amylase, glucose, and cytology) improve the differentiation of PCLs. EUS-FNA of cyst fluid with molecular analysis and nCLE for in vivo endomicroscopy, when indicated (generally PCLs ≥ 2 cm diameter), collectively approach a high (>90%) accuracy for the diagnosis and risk stratification of PCLs [2,3,4,5,9,23].

## Figures and Tables

**Figure 1 diagnostics-13-00065-f001:**
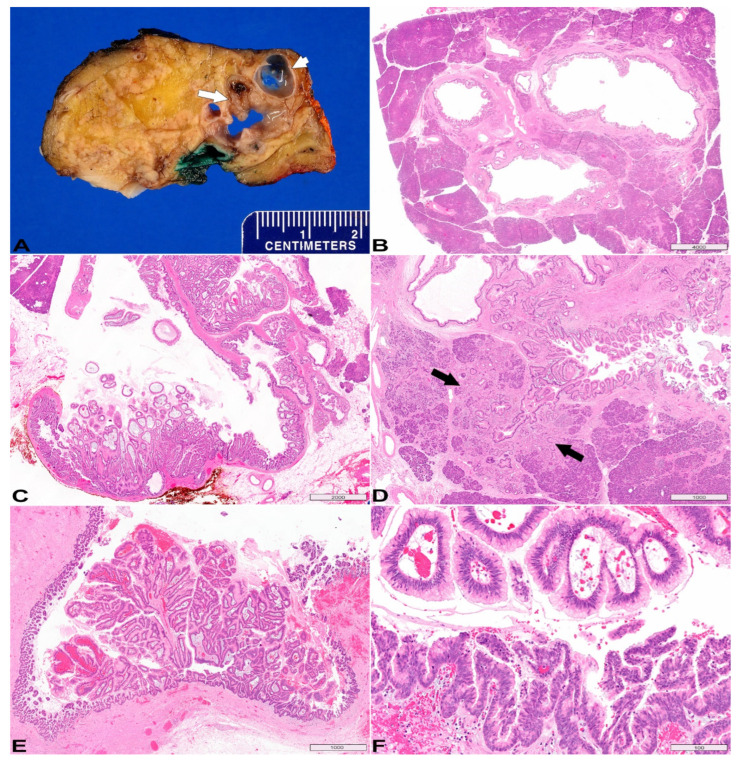
**Intraductal papillary mucinous neoplasm (IPMN).** (**A**) Gross photo of the cut surface of a resected branch-duct IPMN, characterized by multilocular, Swiss cheese appearance; short and long arrows pointing to smooth cyst lining and intraductal papillary growth respectively. (**B**,**C**) Variable (minimal to exuberant) intraductal, papillary epithelial proliferation. (**D**) The pancreatic parenchyma adjacent to IPMN undergoes lobular atrophy (between arrows). (**E**) A main-duct IPMN: the lumen is partially obstructed by papillary/villous neoplastic growth. (**F**) Upper half the image demonstrates low-grade dysplasia (large papillae with abundant cytoplasmic mucin and fibrovascular core, monolayer polarized nuclei), whereas the low half shows high-grade dysplasia (branching and tufting papillae, stratified nuclei with high nuclear cytoplasmic ratio). (**B**–**F**) Hematoxylin & Eosin (H&E) stain, original magnification 5×, 10×, 20×, 20×, 200× respectively. Scale bars, in µm.

**Figure 2 diagnostics-13-00065-f002:**
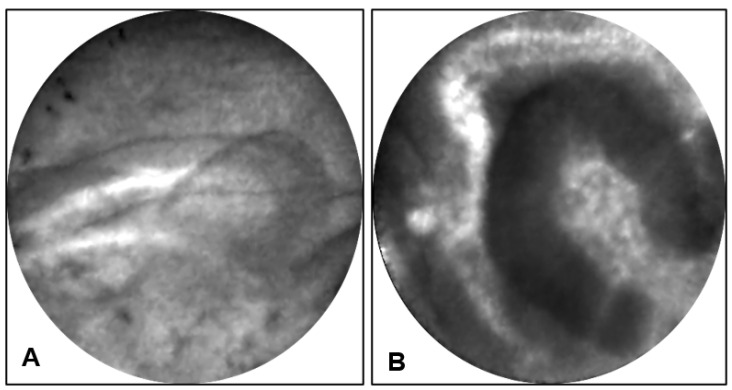
(**A**) EUS-nCLE of BD-IPMN with papillae revealing thin translucent epithelium in a lesion with low-grade dysplasia. (**B**) BD-IPMN revealing large dominant papillary structure with a thick and dark epithelium in a lesion with high-grade dysplasia.

**Figure 3 diagnostics-13-00065-f003:**
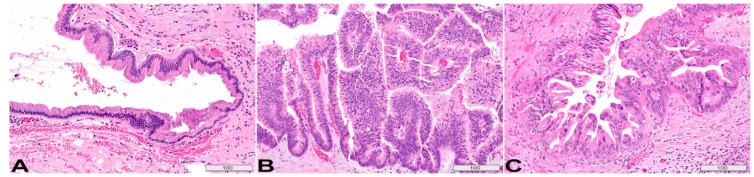
**Histologic subtypes of IPMN.** (**A**) Gastric type resembles gastric foveolar epithelium, composed of tall columnar cells with basally oriented small nuclei and abundant cytoplasmic mucin, typically low-grade dysplasia, commonly seen in branch ducts. (**B**) Intestinal type features hyperchromatic villous papillae with elongated and crowded nuclei, basophilic cytoplasm, often high-grade dysplasia, associated with main duct. (**C**) Pancreaticobiliary type is characterized by amphophilic to eosinophilic cytoplasm, enlarged nuclei with prominent nucleoli, and moderate cytoplasmic mucin. It often shows high-grade dysplasia and typically involves the main duct. All H&E stain, original magnification, 200×. Scale bars, in µm.

**Figure 4 diagnostics-13-00065-f004:**
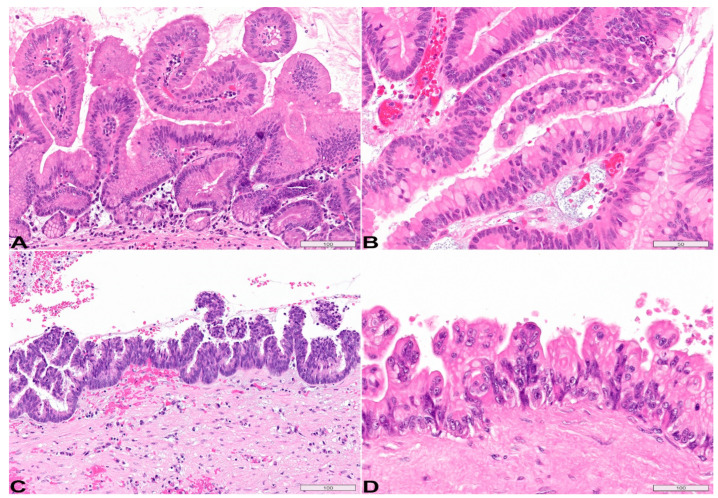
**Two-tiered histologic grading of IPMN: Low-grade and high-grade.** (**A**) showing low-grade epithelium that arranges into tall papillae, but notice the bland cytology and abundant cytoplasmic mucin. (**B**–**D**) High-grade dysplasia feature severe atypia, characterized by papillae with irregular branching and budding, nuclear stratification with loss of polarity, cellular pleomorphism, increased mitoses and variably decreased cytoplasmic mucin. Of note, with increasing dysplasia, the papillae may become shorter/smaller and lose central fibrovascular stromal core. In comparison to the low-grade large thick papillae in (**A**), the high-grade papillae are shorter without stromal core ((**B**), center slender papillae; (**C**,**D**), short stubby papillae). H&E stain, original magnification, (**A**,**C**,**D**) 200×; (**B**) 400×. Scale bars, in µm.

**Figure 5 diagnostics-13-00065-f005:**
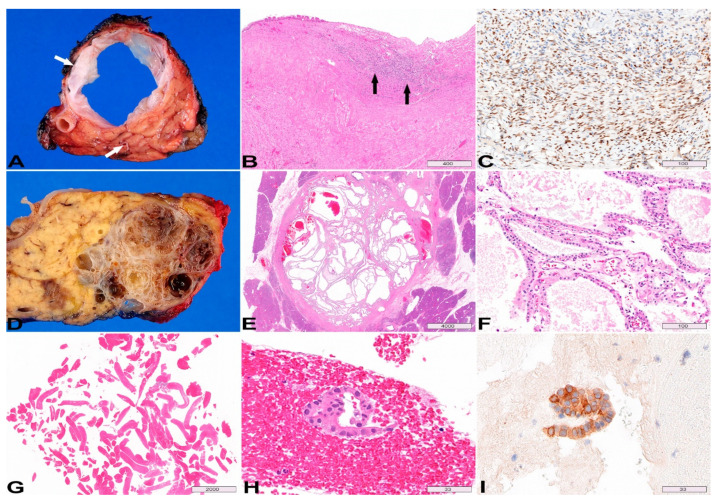
**Mucinous cystic neoplasm (MCN, A–C) and serous cystadenoma (SCA, D–I).** (**A**) Gross photo of the cut surface of a resected MCN, notice the thick fibrous wall (top arrow). It does not communicate with the pancreatic duct (bottom arrow). (**B**) Sections of the MCN shows focal residual epithelium (top left) with large area of denudation. The subepithelial ovarian-type stroma is required for the diagnosis of MCN (arrows). (**C**) Progesterone receptor is positive in the ovarian-type stroma by immunohistochemistry. (**D**,**E**) Cut surfaces of a resected microcystic SCA, characterized by thin, smooth-walled cysts with a sponge or honeycomb appearance. (**F**) Microscopically, the epithelium is flat with clear cuboidal cells and prominent underlying capillary network. (**G**) Microforceps biopsy of SCA typically yields predominantly bloody contents. (**H**) Rare minute strip of non-mucinous epithelial cells is present. (**I**) The neoplastic epithelium is immunoreactive for inhibin, which can be helpful for the diagnosis of SCN in small biopsy or cytology specimens. (**B**,**E**–**H**), H&E stain, (**C**), progesterone receptor immunostain, (**I**), inhibin immunostain; original magnification, (**B**) 50×, (**C**) 200×, (**E**) 5×, (**F**) 200×, (**G**) 10×, (**H**,**I**) 600×. Scale bars, in µm.

**Figure 6 diagnostics-13-00065-f006:**
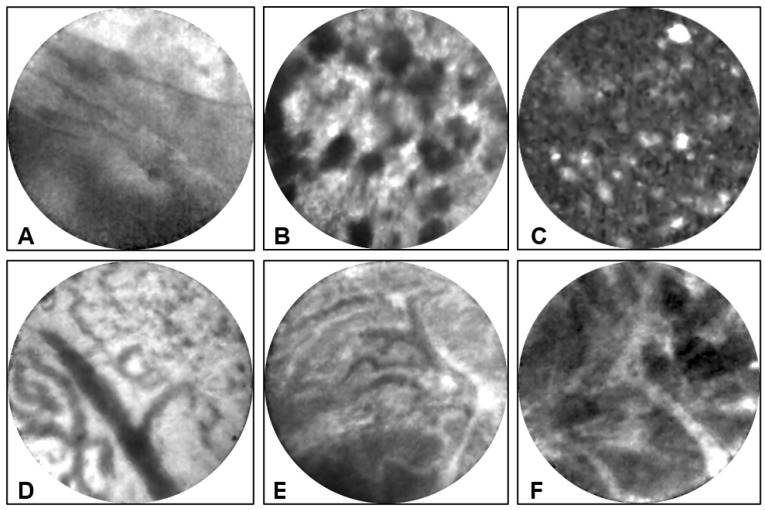
**EUS-nCLE images of PCLs.** (**A**) A **mucinous cystic neoplasm** revealing layering epithelial bands with thin epithelium in a lesion with low-grade dysplasia. (**B**,**C**) A **pseudocyst** revealing dark clumps of inflammatory debris (**B**) and a dark background (**C**) due to the absence of a vascular interstitium. (**D**,**E**) A **serous cystadenoma** demonstrating a fern-pattern of capillary network: The capillaries are packed with red blood cells (**D**) and partial contrast penetration highlighting the flow of red blood cells within the capillary network (**E**). (**F**) A **cystic neuroendocrine tumor** showing clusters of cells in geometric shapes (dark regions) separated by the stroma of the lesion.

**Figure 7 diagnostics-13-00065-f007:**
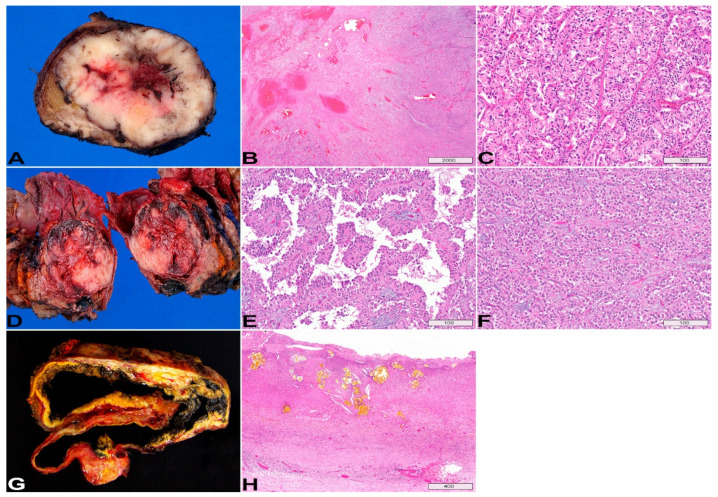
Cystic neuroendocrine tumor (cNET, **A**–**C**), solid pseudopapillary neoplasm (SPN, **D**–**F**), and pseudocyst (**G**,**H**). (**A**) Gross photo showing an early cNET that has a solid cut surface with central hemorrhage/cystic degeneration. (**B**) Corresponding histology image showing hemorrhage with fibrosis. (**C**) cNET is composed of nests, trabeculae, insular organization of the neoplastic cells, separated by thin vascular fibrous septa. (**D**) Gross photo of the cut surface of a resected SPN, multifocal hemorrhage, necrosis, and cystic degeneration are apparent. (**E**) Pseudopapillary structures due to neoplastic perivascular growth and the degeneration of the tumor cells in between the papillae from lack of blood supply. (**F**) As the name implies, there are also solid areas in SPN, which resembles pancreatic NET on both histology and endomicroscopy. (**G**) Gross photo of the cut surface of a resected pseudocyst that is located in between the pancreatic tail and the spleen. (**H**) A thick fibrotic wall without epithelial lining is seen in this chronic pseudocyst. The cyst wall consists of hematoidin pigment, inflammation, granulation tissue and fibrosis. (**B**,**C**,**E**,**F**,**H**), H&E stain, original magnification, (**B**), 10×; (**C**,**E**,**F**), 200×; (**H**), 50×. Scale bars, in µm.

**Table 1 diagnostics-13-00065-t001:** Comparison of clinicopathological features of pancreatic cystic lesions.

	IPMN	MCN	SCA	cNET	SPN	Pseudocyst
**Incidence ***	45%	16%	16%	5%	3%	-
**Mean Age (year)**	65	45	62	-	25–30	-
**Sex**	M > F	F >> M	F > M	-	F >> M	-
**Mucinous**	Yes	Yes	No	No	No	No
**Ductal communication**	Yes	No	No	No	No	No
**Mean size**	Variable	6 cm	4 cm	Variable	8 cm	Variable
**Most common location**	Head	Tail/ body	Body/tail	Variable	Tail	Usually outside pancreas; exophytic
**Cyst fluid**	High viscosity, high amylase, CEA > 192 ng/mL	High viscosity, CEA > 192 ng/mL	Low viscosity, low CEA < 192 ng/mL	Low viscosity, low CEA < 192 ng/mL	-	Low viscosity, high amylase
**Macroscopic features**	Variable wall thickness, smooth to papillary lining epithelium	Thick wall ≥ 3 mm, unilocular or multilocular with few septa	Thin wall < 3 mm, smooth cyst lining, microcystic (honey comb) > macro/oligocystic > solid	Unilocular, hemorrhage, serous fluid	Solid with cystic spaces, hemorrhage, necrosis	Unilocular, no lining epithelium, dark cloudy fluid, ultimately thick fibrous wall
**EUS-nCLE**	Papillary projections with outer epithelium and inner vascular core	Horizon-type epithelial bands without papillae conformation	Distinct vascular pattern–an intricate fern pattern of capillary networks	Dark clusters (trabeculae) of cells in cords or nests separated by stroma	Dark clusters (trabeculae) of cells in cords or nests separated by stroma	Clumps of inflammatory cells. Dark background due to the absence of epithelium and associated vascular interstitium
**Histology**	Papillary mucinous epithelium	Mucinous epithelium, ovarian-type stroma	Flat serous epithelium, clear cytoplasm, subepithelial capillary network	Nests/trabeculae of cells separated by fibrous bands	Solid nested areas and pseudopapillary structures	No cyst epithelium, inflammatory/fibrotic wall
**IHC**	Gastric-type: MUC5AC+; Intestinal-type: MUC2+/CDX2+; Pancreatobiliary type: MUC1/MUC6+	PR > ER+, inhibin+	Inhibin+	synaptophysin+, chromogranin+	Beta-catenin (nuclear), SOX11+, TFE3+	-
**Molecular** **alteration**	*MAPK/GNAS*, *RNF43*. Advanced neoplasia: *TP53, SMAD, CDKN2A, mTOR*	*MAPK, RNF43*. Advanced neoplasia: *TP53, SMAD, CDKN2A, mTOR*	*VHL*	*MEN1,* LOH	*CTNNB1*	-

Data sources [1,3,9,13,14,52,66,72]. * Based on surgical series [10,11,12]. Abbreviations: CEA = carcinoembryonic antigen, ER = estrogen receptor, EUS-nCLE = EUS-guided needle based confocal laser endomicroscopy, cNET = cystic neuroendocrine tumor, IHC = Immunohistochemistry, IPMN = intraductal papillary mucinous neoplasm, LOH = loss of heterozygosity, MCN = mucinous cystic neoplasm, PR = progesterone receptor, SCA = serous cystadenoma, SPN = solid pseudopapillary neoplasm. MAPK genes include *KRAS, BRAF, and NRAS*; mTOR genes include *PIK3CA, PTEN, AKT1*.

## Data Availability

Not applicable.

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
