# Peer review of "Pancreatic Cystic Lesions: A Focused Review on Cyst Clinicopathological Features and Advanced Diagnostics"

_diagnostics, 2022, doi:10.3390/diagnostics13010065_

Round 1

Reviewer 1 Report

The review manuscript ”The Ultimate Approach for Evaluating Pancreatic Cystic Lesions: Macroscopic, Endomicroscopic, Molecular, and Histologic Correlation” makes a well written summary of the methods available to diagnose different pancreatic cysts. There are some concerns which need to be addressed:

1. Please indicate image magnification by inserting scale bar, not by indicating magnification.

2. You should also include mass-spectrometry as a method to identify pancreatic cysts and reference the article Jabbar KS et al., 2018: Highly Accurate Identification of Cystic Precursor Lesions of Pancreatic Cancer Through Targeted Mass Spectrometry: A Phase IIc Diagnostic Study, doi: 10.1200/JCO.2017.73.7288.

3. The summary should contain a paragraph about why these lesions appear. Why are some more common in females, for example? The results section contains information about mutated genes, but what are the implications for mutations in these genes? A review should contain some comments on why and not only what it looks like.

Author Response

Dear Editors,

We sincerely appreciate the positive feedbacks from the reviewers. We have made every effort to thoroughly address the Reviewers’ helpful comments and suggestions. Please see our responses itemized and bolded below.

The review manuscript ”The Ultimate Approach for Evaluating Pancreatic Cystic Lesions: Macroscopic, Endomicroscopic, Molecular, and Histologic Correlation” makes a well written summary of the methods available to diagnose different pancreatic cysts. There are some concerns which need to be addressed: 

  1. Please indicate image magnification by inserting scale bar, not by indicating magnification.

This was done.

  1. You should also include mass-spectrometry as a method to identify pancreatic cysts and reference the article Jabbar KS et al., 2018: Highly Accurate Identification of Cystic Precursor Lesions of Pancreatic Cancer Through Targeted Mass Spectrometry: A Phase IIc Diagnostic Study, doi: 10.1200/JCO.2017.73.7288.

A new section “Advanced and Emerging Diagnostic Tools for Pancreatic Cystic Lesions” is added before the summary, where mass spectrometry is discussed. 

  1. The summary should contain a paragraph about why these lesions appear. Why are some more common in females, for example? The results section contains information about mutated genes, but what are the implications for mutations in these genes? A review should contain some comments on why and not only what it looks like.

These are great suggestions and the text was revised as suggested. A paragraph discussing the etiology of PCLs and why some are common in females are added in the beginning of the summary. The implications of the gene mutations are discussed in the 3rd paragraph in the new section “Advanced and Emerging Diagnostic Tools for Pancreatic Cystic Lesions”.

Reviewer 2 Report

The current manuscript covers very well, cystic pancreatic tumors from several points of view and emphasizes the potential of using a combination of various techniques for tumor characterization. The authors should decide if this review is a pathology review or if it involves other techniques. 

I have some recommendations:

1. The introduction should be more thorough and should better introduce the available guidelines and recommendations, as well as the objective of the study

2. I would not call it a "multidisciplinary approach", since no imaging techniques are involved, such as MRI or EUS (either contrast-enhanced or not) since they are the first imaging techniques recommended before surgical resection. Thus I recommend reconsidering the title and where this term is used in the text

3. I recommend either expanding the summary or inserting another chapter focusing on new techniques that may aid the diagnosis. Is there any future for CLE in the setting of cystic lesions? Should guidelines recommend its use? Is it time for standardization?  CLE molecular imaging should be consider?

4. Is EUS through the needle biopsy forceps helpful for these lesions? Should it be recommended before surgery? Is it helpful for the pathology setting?

Author Response

Dear Editors,

We sincerely appreciate the positive feedbacks from the reviewers. We have made every effort to thoroughly address the Reviewers’ helpful comments and suggestions. Please see our responses itemized and bolded below.

The current manuscript covers very well, cystic pancreatic tumors from several points of view and emphasizes the potential of using a combination of various techniques for tumor characterization. The authors should decide if this review is a pathology review or if it involves other techniques. 

Thank you for pointing this out. The title and text of the manuscript have been revised to align with the pathology-heavy nature of this review. The new title isPancreatic Cystic Lesions: A Focused Review on Cyst Clinicopathological Features and Advanced Diagnostics”.

I have some recommendations:

  1. The introduction should be more thorough and should better introduce the available guidelines and recommendations, as well as the objective of the study

In the introduction, we have added the guidelines that our institution follows as well as the objective of the study as suggested. A thorough review of the available guidelines and recommendations is out of the scope of this review, and the readers are referred to other dedicated recent reviews including one in this special issue of pancreatic cysts.

  1. I would not call it a "multidisciplinary approach", since no imaging techniques are involved, such as MRI or EUS (either contrast-enhanced or not) since they are the first imaging techniques recommended before surgical resection. Thus I recommend reconsidering the title and where this term is used in the text.

We agree with the reviewer and revised the title/text accordingly as mentioned above.

  1. I recommend either expanding the summary or inserting another chapter focusing on new techniques that may aid the diagnosis. Is there any future for CLE in the setting of cystic lesions? Should guidelines recommend its use? Is it time for standardization?  CLE molecular imaging should be consider?

A new section “Advanced and Emerging Diagnostic Tools for Pancreatic Cystic Lesions” is added before the summary, where nCLE, cyst fluid molecular analysis, EUS through-the-needle biopsy, mass spectrometry, and OCT are discussed. To the best of our knowledge, data on CLE molecular imaging of human pancreas are not available, so this is not included in the review.  

  1. Is EUS through the needle biopsy forceps helpful for these lesions? Should it be recommended before surgery? Is it helpful for the pathology setting?

One paragraph dedicated to EUS through-the-needle biopsy is added to the new section “Advanced and Emerging Diagnostic Tools for Pancreatic Cystic Lesions”.

Round 2

Reviewer 2 Report

I appreciate following my recommendations and I endorse the manuscript for publication.